# Two distinct immunopathological profiles in autopsy lungs of COVID-19

Ronny Nienhold[1,10], Yari Ciani [2,10], Viktor H. Koelzer [3,4,10], Alexandar Tzankov [5], Jasmin D. Haslbauer[5], Thomas Menter [5], Nathalie Schwab[1], Maurice Henkel [1], Angela Frank[1], Veronika Zsikla[1], Niels Willi[1], Werner Kempf[6], Thomas Hoyler[7], Mattia Barbareschi [8], Holger Moch[3], Markus Tolnay[5], Gieri Cathomas[1], Francesca Demichelis [2,9,11], Tobias Junt[7,11] & Kirsten D. Mertz [1,11✉]

Coronavirus Disease 19 (COVID-19) is a respiratory disease caused by severe acute respiratory syndrome coronavirus 2 (SARS-CoV-2), which has grown to a worldwide pandemic with substantial mortality. Immune mediated damage has been proposed as a pathogenic factor, but immune responses in lungs of COVID-19 patients remain poorly characterized. Here we show transcriptomic, histologic and cellular profiles of post mortem COVID-19 ($n = 34$ tissues from 16 patients) and normal lung tissues ($n = 9$ tissues from 6 patients). Two distinct immunopathological reaction patterns of lethal COVID-19 are identified. One pattern shows high local expression of interferon stimulated genes (ISG$^{high}$) and cytokines, high viral loads and limited pulmonary damage, the other pattern shows severely damaged lungs, low ISGs (ISG$^{low}$), low viral loads and abundant infiltrating activated CD8$^+$ T cells and macrophages. ISG$^{high}$ patients die significantly earlier after hospitalization than ISG$^{low}$ patients. Our study may point to distinct stages of progression of COVID-19 lung disease and highlights the need for peripheral blood biomarkers that inform about patient lung status and guide treatment.

[1] Institute of Pathology, Cantonal Hospital Baselland, Liestal, Switzerland. [2] Laboratory of Computational and Functional Oncology, Department for Cellular, Computational and Integrative Biology – CIBIO, University of Trento, Trento, Italy. [3] Department of Pathology and Molecular Pathology, University Hospital Zurich, Zurich, Switzerland. [4] Department of Oncology and Nuffield Department of Medicine, University of Oxford, Oxford, UK. [5] Pathology, Institute of Medical Genetics and Pathology, University Hospital Basel, Basel, Switzerland. [6] Kempf und Pfaltz Histologische Diagnostik, Zurich, Switzerland. [7] Novartis Institutes for BioMedical Research (NIBR), Basel, Switzerland. [8] Anatomia ed Istologia Patologica, Ospedale S. Chiara di Trento, Trento, Italy. [9] Caryl and Israel Englander Institute for Precision Medicine, Institute for Computational Biomedicine, New York Presbyterian Hospital, Weill Cornell Medicine, New York, NY, USA. [10] These authors contributed equally: Ronny Nienhold, Yari Ciani, Viktor H. Koelzer. [11] These authors jointly supervised this work: Francesca Demichelis, Tobias Junt, Kirsten D. Mertz. ✉ email: kirsten.mertz@ksbl.ch

Coronavirus Disease 19 (COVID-19) is a pandemic respiratory disease with 2–3% lethality and a particularly severe course in males, patients with cardiovascular comorbidities, and in the elderly[1, 2]. Lymphopenia, high levels of pro-inflammatory cytokines in the circulation[3], and phenotypic changes of pro-inflammatory macrophages in bronchoalveolar lavages (BALs)[4] in severe patients have led to the notion that the immune response against the causative virus severe acute respiratory syndrome coronavirus 2 (SARS-CoV-2) may contribute to devastating end-organ damage[5]. Since patients with severe COVID-19 may develop acute respiratory distress syndrome (ARDS) and many patients die from respiratory failure with diffuse alveolar damage[6], it is critical to understand the immunological profiles in the lungs of these patients.

In this work, we use histologic and transcriptomic analyses of post mortem lung tissues in a cohort of patients where the cause of death was respiratory failure. We describe two distinct immunological and cellular profiles in the lungs of these patients, defined by their differential expression of interferon-stimulated genes (ISGs) and immune infiltration patterns. ISG subgroups strongly differ in regards to the characteristics and the extent of pulmonary damage, pulmonary viral loads, immune infiltration, and time from hospitalization to death. These data highlight two distinct patterns of immune pathology of pulmonary COVID-19 and may give insight into the natural progression of COVID-19 in the lungs.

## Results

**Two patterns of gene expression in COVID-19 autopsy lungs.** Here we analyzed 34 post mortem lung samples from 16 deceased COVID-19 patients and 9 post mortem lung samples from 6 patients, who died from non-infectious causes (Table 1 and Supplementary Table 1). The primary cause of death in all patients of this cohort was respiratory failure, sometimes multi-organ failure including failure of the respiratory system. Since lung samples from the same patients did not always appear morphologically uniform, all lung specimens were subjected to differential gene expression analysis based on a commercially available targeted next-generation sequencing (NGS) assay (OIRRA) designed for the quantification of immune cell and inflammatory transcripts (Supplementary Table 2). Among the 398 genes investigated, we identified 68 upregulated and 30 downregulated genes in COVID-19 infected lungs compared to normal tissue (Fig. 1a, b and Supplementary Table 3), and a PCA analysis showed segregation of COVID-19 patients in two well-defined clusters that showed distinct association with viral load (Fig. 1a, c).

Using a consensus of 30 different indices[7], we identified three groups of samples defined by distinct expression levels of the deregulated genes by K-means clustering (Fig. 1a). Clusters 1 (50% of samples) and 2 (41%) contained COVID-19 samples while cluster 3 contained all normal lung samples as well as three COVID-19 samples (9%). To understand why the majority of COVID-19 lung tissues segregated into defined clusters 1 or 2, we undertook a gene ontology analysis. We identified ISGs as a key upregulated pathway in COVID-19 autopsy lungs (Table 2), which was differentially represented in clusters 1 and 2, respectively (Fig. 1d). Identification of an ISG[high] cluster (Cluster 1, ISG[high]) was surprising since SARS-CoV-2 was recently proposed to lead to limited ISG induction, yet only in comparison to other respiratory viruses[8]. Our data suggest that autopsy lungs of COVID-19 patients, who died from respiratory failure, showed two different gene expression patterns with different levels of ISG activation (ISG[high] and ISG[low]).

**Clinical differences of COVID-19 patients with an ISG[high] versus ISG[low] lung profile.** Patients, whose lung samples all segregated in the ISG[high] or in the ISG[low] subgroups, were called ISG[high] and ISG[low] patients. To investigate whether there were clinical differences between these two patient groups, we compared their clinical and epidemiological information. The majority of COVID-19 patients in our cohort (81%) were male and the average body mass index (BMI) was 31.4. kg/m$^2$. The patient-level analysis revealed no correlation between sex or BMI with the ISG patterns (Fig. 1a). All patients in our cohort died from respiratory failure or multi-organ failure including failure of the respiratory system, independent of ISG subgrouping (Supplementary Table 1). When we analyzed comorbidities and autopsy findings, we found that 5 out of 7 (71%) ISG[low] patients, but none of the ISG[high] patients had an autoptic finding of a thromboembolic event in the lungs and/or disseminated

### Table 1 Clinical data of all patients.

| Characteristics | N or x | % or range |
|---|---|---|
| COVID-19 cohort (16 patients) | | |
| Time between symptoms and death | 7.4 d | 0–20 d |
| Hospitalization | 5.6 d | 0–13 d |
| Post mortem interval | 28.4 h | 11–67 h |
| Age | 75 y | 53–96 y |
| Sex | M 13: F 3 | |
| Comorbidities | | |
| Hypertension | 16 | 100% |
| Cardiovascular disease | 11 | 68% |
| (Pre-)adipositas | 12 | 75% |
| Diabetes | 6 | 37.5% |
| Initial clinical presentation | | |
| Cough | 13 | 81.25% |
| Fever | 12 | 75% |
| Dyspnea | 6 | 37.5% |
| Renal failure | 6 | 37.5% |
| Laboratory results | | |
| Interleukin-6 (IL6) | 5774.72 ng/l | 159.00–35,152.00 ng/l |
| C-reactive protein (CRP) | 216.36 mg/l | 36.00–512.00 mg/l |
| Ferritin | 18,037.71 µg/l | 1025.00–228,225.00 µg/l |
| Procalcitonin (PCT) | 1996.92 µg/l | 0.47–5300.00 µg/l |
| Lactate dehydrogenase (LDH) | 870.42 U/l | 256.00–5267.00 U/l |
| Troponin T (cTnT) | 54.65 ng/l | 2.07–126.00 ng/l |
| Treatment | | |
| Hydroxychloroquine | 10 | 62.5% |
| Lopinavir/Ritonavir | 5 | 31.25% |
| Antibiotics | 12 | 75% |
| ACTEMRA (Tocilizumab) | 5 | 31.25% |
| Listed laboratory results correspond to the highest (LDH, cTnT), latest (IL6, ferritin, PCT), or last value before administration of ACTEMRA/Tocilizumab (CRP) | | |
| Control cohort (6 patients) | | |
| Post mortem interval | 31.4 h | 25–46 h |
| Age | 81.3 y | 63–104 y |
| Sex | M 5: F 1 | |
| Cohort of patients with other infections (4 patients) | | |
| Post mortem interval | 31.5 h | 12–79 h |
| Age | 79.5 y | 58–81 y |
| Sex | M 2: F 2 | |
| Patients were suffering from other infections of the lung (bacterial or viral pneumonia). Detailed analysis of individual pathogens is shown in Fig. 2 | | |

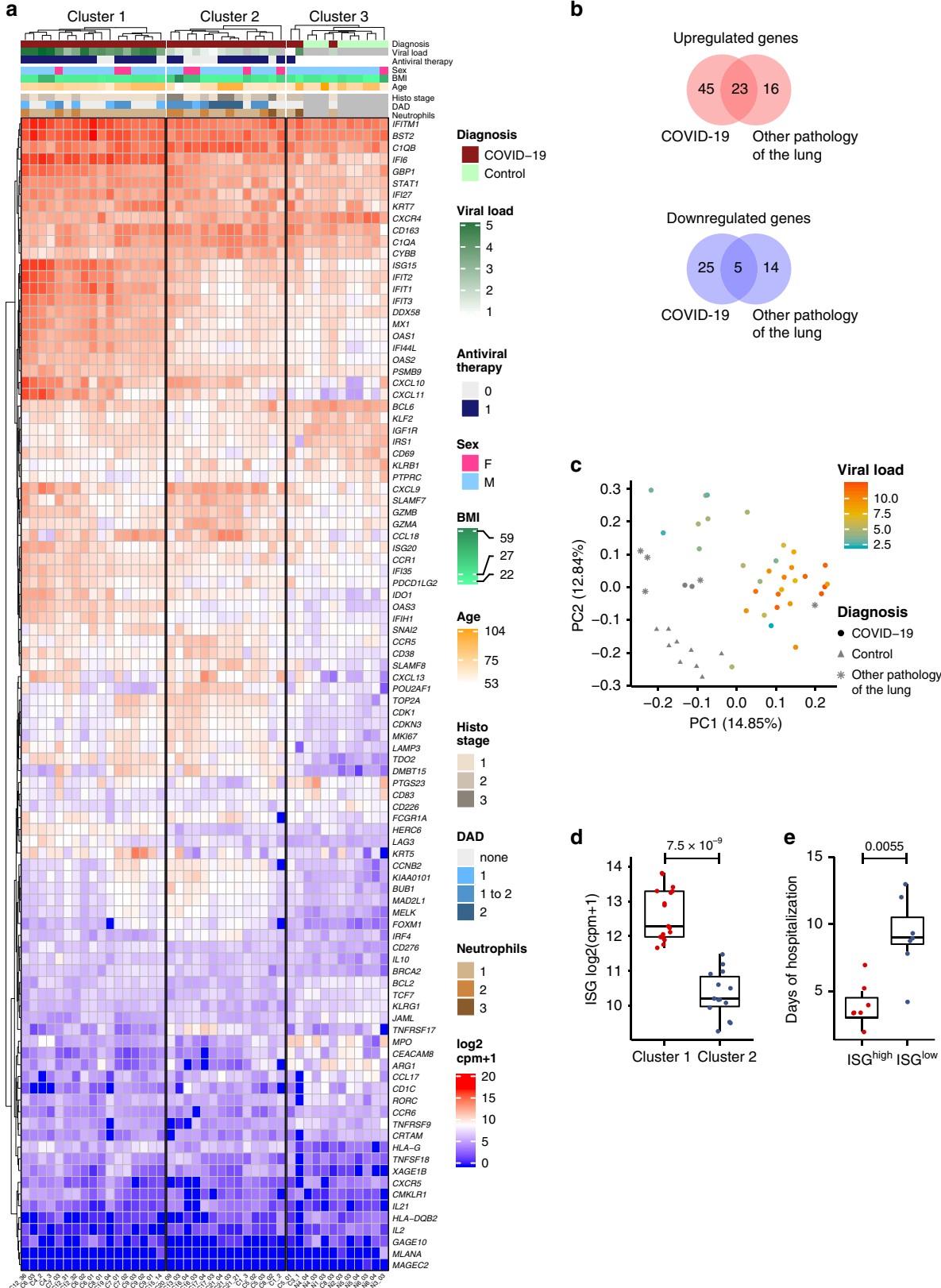

intravascular dissemination (DIC) indicating abnormally activated blood coagulation (hypercoagulability) exclusively in ISG$^{low}$ patients (Supplementary Table 1). COVID-19-associated coagulopathy was unlikely to contribute to the rapid exacerbation of pulmonary COVID-19 since this diagnostic criterion was only observed in patients displaying the ISG$^{low}$ phenotype. Most

notably, when we compared the hospitalization time between the patients with different ISG lung profiles, we found a significantly longer hospitalization time for ISG$^{low}$ patients from admission to death compared to ISG$^{high}$ patients (Fig. 1e). Of note, we did not find differences in the anamnestic onset of disease symptoms and time from positive testing for COVID-19 infection by

**Fig. 1 ISG^high and ISG^low are two gene expression profiles in COVID-19 autopsy lungs. a** Heatmap showing *K*-means clustering of COVID-19 and normal lung samples based on expression levels of deregulated genes in COVID-19 versus normal lungs. **b** Comparison of upregulated and downregulated genes in lung samples from COVID-19 patients, normal lung samples, and samples from other infectious lung pathologies. **c** Principal component analysis (PCA) of COVID-19 and non-COVID-19 lung samples reveals segregation in two distinct groups based on diagnosis and viral load. **d** ISG signature expression in clusters 1 and 2 of COVID-19 lungs defines two profiles of COVID-19 autopsy lungs termed ISG^high and ISG^low. Study patients with unambiguous sample segregation in either Cluster 1 or 2 were assigned the corresponding ISG activation label ISG^high and ISG^low, respectively ($n = 31$ independent samples). **e** Hospitalization time in ISG^high patients versus ISG^low patients ($n = 14$ independent samples). ISG^high samples, red; ISG^low samples, blue. Box-plots elements indicate the median (center line), upper and lower quartiles (box limits). Whiskers extend to the most extreme value included in 1.5× interquartile range. Groups were compared using a two-sided Wilcoxon rank-sum test.

---

**Table 2 Gene ontology enrichment analysis of genes upregulated in COVID-19 samples.**

| ID | Description | GeneRatio | BgRatio | *p*-value | *p*.adjust |
|---|---|---|---|---|---|
| GO:0034340 | Response to type I interferon | 17/66 | 30/379 | 3.70E−07 | 0.000158 |
| GO:0060337 | Type I interferon signaling pathway | 17/66 | 30/379 | 3.70E−07 | 0.000158 |
| GO:0071357 | Cellular response to type I interferon | 17/66 | 30/379 | 3.70E−07 | 0.000158 |
| GO:0051607 | Defense response to virus | 21/66 | 44/379 | 5.53E−07 | 0.000177 |
| GO:0098542 | Defense response to other organism | 25/66 | 71/379 | 3.85E−05 | 0.009888 |
| GO:0009615 | Response to virus | 21/66 | 57/379 | 9.79E−05 | 0.020960 |
| GO:0045069 | Regulation of viral genome replication | 10/66 | 18/379 | 1.87E−04 | 0.034381 |
| GO:1903900 | Regulation of viral life cycle | 11/66 | 22/379 | 2.95E−04 | 0.042931 |
| GO:0045071 | Negative regulation of viral genome replication | 8/66 | 13/379 | 3.57E−04 | 0.042931 |
| GO:1903901 | Negative regulation of viral life cycle | 8/66 | 13/379 | 3.57E−04 | 0.042931 |
| GO:0048525 | Negative regulation of viral process | 9/66 | 16/379 | 3.68E−04 | 0.042931 |
| GO:0051603 | Proteolysis involved in cellular protein catabolic process | 7/66 | 11/379 | 6.71E−04 | 0.071789 |
| GO:0050792 | Regulation of viral process | 12/66 | 28/379 | 9.17E−04 | 0.088399 |
| GO:0019079 | Viral genome replication | 10/66 | 21/379 | 9.64E−04 | 0.088399 |
| GO:0043901 | Negative regulation of multi-organism process | 9/66 | 18/379 | 1.16E−03 | 0.099123 |

Significance calculated with hypergeometric test, fdr corrected.

---

nasopharyngeal swab to hospitalization or death between ISG^high and ISG^low patients. To exclude bacterial and viral super-infections as a confounder of clinical course, we performed whole-genome sequencing on all samples to detect bacterial and/or viral DNA. Bacterial superinfections were found in three lung tissue samples, in 3/16 COVID-19 patients, that were equally distributed among the different groups (Fig. 2a–e). Based on the limited number of samples with evidence of bacterial super-infection, there was no correlation with clinical subgroups or the duration of the disease.

Taken together, expression of the ISG^high profile in COVID-19 lungs is associated with early lethal outcome, and the ISG^low profile is associated with coagulopathies and later lethal outcomes. Although autopsy studies do not allow for long-itudinal analyses, our observation of distinct immunological patterns of COVID-19 lungs at different times after hospitalization is very suggestive of a natural disease course of COVID-19 in lungs from an ISG^high profile to an ISG^low profile, consistent with a longitudinal study in peripheral blood showing that ISG expression was high in early COVID-19 and declined later[9].

**Immune microenvironment characteristics of the ISG^high and ISG^low COVID-19 lung profiles.** In line with a recent study showing a correlation of ISG expression and viral load in naso-pharyngeal swabs[10], expression of ISGs was positively correlated with pulmonary viral load (Fig. 3a), and immunohistochemical staining confirmed the presence of SARS-CoV-2 nucleocapsid protein in ISG^high lungs, mainly localized to pneumocytes (Fig. 3b).

SARS-CoV-2 induces a strong antiviral immune response. Therefore, we analyzed frequencies of specific immune cells in the lungs by computational image analysis. T cells (CD3^+) of the CD4^+ and CD8^+ lineages, B cells (CD20^+), and macrophages (CD68^+) were selectively enriched in lung tissues from ISG^low patients (Figs. 3c, d and 4a, b). A strong enrichment for CD68^+ and CD163^+ monocytes in lung tissue was observed with a spatial correlation of stains for both markers indicating co-expression. Since circulating monocytes in COVID-19 patients co-express CD68 and CD163, it was not surprising that CD68 and CD163 expression in lungs followed a similar pattern[11] (Fig. 3c, d). Surprisingly, CD123^+ plasmacytoid dendritic cells (pDCs) did not show elevated frequencies in ISG^high lungs (Fig. 4a, b), and our analysis did not allow us to identify the upstream trigger of ISGs in lungs.

Innate cytokines have been proposed to contribute to an adverse outcome of COVID-19[12] and cytokines are highly expressed in BALs of COVID-19 patients[4]. Therefore, we investigated the expression of a pro-inflammatory cytokine signature (*TNF*, *IL1B*, *IL6*, *CCL2*, *IFNA17*, *IFNB1*, *CXCL9*, *CXCL10*, *CXCL11*) in lung samples from lethal COVID-19, which contains genes that are upregulated in plasma and/or BAL of severe COVID-19 patients[4, 12], and type I interferons, which are deregulated in COVID-19[8]. This pro-inflammatory gene signature was significantly enriched in the ISG^high subset ($p = 0.0061$) (Fig. 5a). Activated CD8^+ T cells are essential for the elimination of coronaviruses[13, 14]. Therefore, we defined and investigated an activated cytotoxic T cell signature (*CD38*, *GZMA*, *GZMB*, *CCR5*) consisting of CD8^+ T cell markers that are associated with severe COVID-19 infection[15, 16], and we found that it was inversely correlated to viral counts, particularly in ISG^low cases (Fig. 5b). This suggests that activated CD8^+ T cells may indeed contribute to the elimination of SARS-CoV-2 in the lungs.

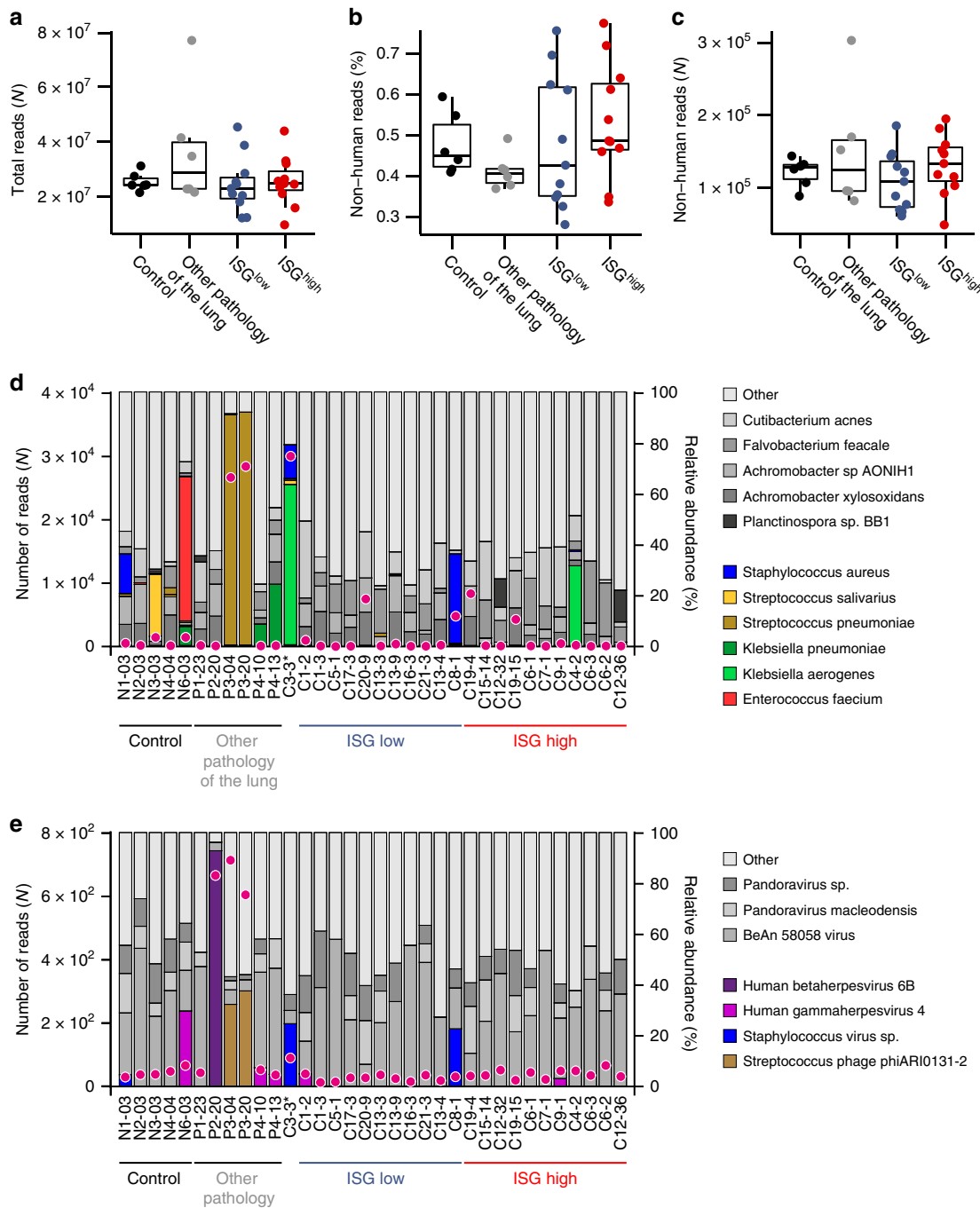

**Fig. 2 Co-infections in COVID-19 lungs identified by WGS metagenomics.** No differences in co-infections in ISG^high and ISG^low COVID-19 lungs identified by WGS metagenomics. **a** Total number of reads generated for each sample. **b** Percentage of reads and **c** absolute numbers of reads not mapping to the human genome (GRCh37 hg19) ($n = 34$ independent samples). Box-plots elements indicate the median (center line), upper and lower quartiles (box limits). Whiskers extend to the most extreme value included in 1.5× interquartile range. **d** Bacterial and **e** viral co-infections across lung samples, WGS metagenomic analysis. Purple dots, numbers of reads sufficient for identification of non-human species. Samples are ordered by increasing the SARS-CoV-2 viral load in both the ISG^low and the ISG^high group. Stacked bars, the relative abundance of the most common species. Gray bars represent frequent species, colored bars show pathogenic species. *One COVID-19 patient (C3) clustered in the normal control group. ISG^high samples, red; ISG^low samples, blue.

Taken together these data show that COVID-19 autopsy lungs with an ISG^high profile show high virus titers, high local expression of innate cytokines, and weak immune cell infiltration, while COVID-19 lungs with an ISG^low profile show low virus titers, lower expression of innate cytokines, and strong immune cell infiltration. This pattern could indicate expression of the lung ISG^high profile at an earlier, innate disease stage, i.e., at a time when the virus is not yet controlled, and expression of the ISG^low profile at a later disease stage, i.e., after T cell priming.

**The ISG^high and ISG^low lung immunoprofiles correlate with morphological changes.** To investigate the potential immunological causes for lung damage in COVID-19, we studied whether

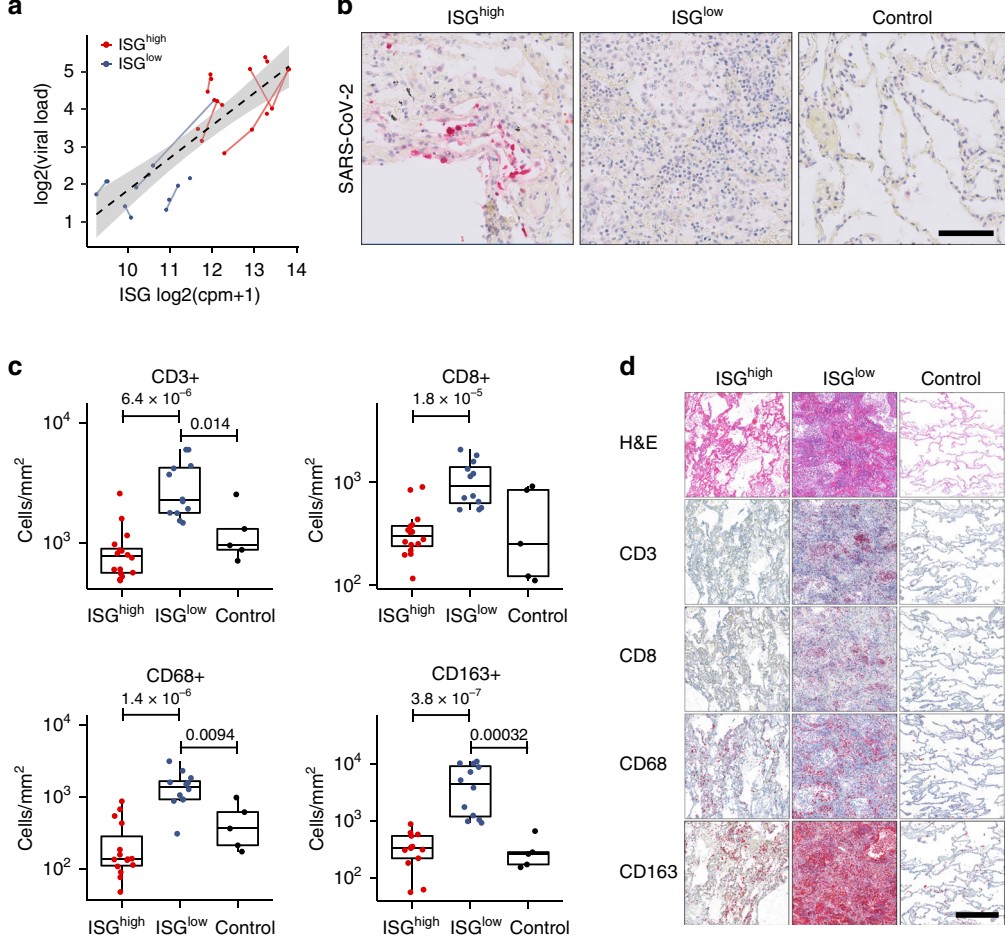

**Fig. 3 Virological and cellular characteristics of the ISG^high and ISG^low COVID-19 lung profiles. a** Correlation of viral load and ISG expression in COVID-19 lungs. Solid lines connect sample points from the same patient. The dotted line shows a regression for all samples, and the gray area delimits the 95% confidence intervals around it (Pearson's correlation = 0.83, adjusted $R$-squared = 0.68, $p$-value = 1.66e−08). **b** Representative immunohistochemistry for SARS-CoV-2 on ISG^high and ISG^low COVID-19 lung samples and controls. Size bar 100 μm. At least two different tissue blocks from different areas of the lungs were evaluated for each case. **c** Frequencies of immune cells on ISG^high and ISG^low COVID-19 lung sections and controls. ($n$ = 33 for CD3 and CD8, $n$ = 32 for CD68, $n$ = 30 for CD163). Box-plots elements indicate the median (center line), upper, and lower quartiles (box limits). Whiskers extend to the most extreme value included in 1.5× interquartile range. Groups were compared using a two-sided Wilcoxon rank-sum test. **d** Representative H&E stains and immunohistochemistry (CD3, CD8, CD68, CD163) of ISG^high and ISG^low COVID-19 lungs and controls, size bar 500 μm. At least two different tissue blocks from different areas of the lungs were evaluated for each case. ISG^high samples, red; ISG^low samples, blue.

ISG profiles in COVID-19 post mortem lungs were associated with specific histomorphological features of fatal COVID-19. Diffuse alveolar damage (DAD) was mostly found in ISG^low patients (Fig. 5c), but intra-alveolar hemorrhage (IAH) was not associated with lung ISG status.

As some cytokines were proposed to contribute to the decline of COVID-19 patients, we analyzed whether expression of the above-defined pro-inflammatory cytokine signature was associated with IAH, but this was not the case (Fig. 5d). However, within this cytokine signature, we identified co-regulated subgroups (*IL1B/IL6/TNF*, *IFNB1/IFNA17*, *CCL2/CXCL9/CXCL19/CXCL11*) (Fig. 5e). Of these, the *CXCL9/10/11* sub-signature was positively associated with IAH (Fig. 5f–i). This is in line with observations that these chemokines compromise endothelial integrity via *CXCR3*[17], and that *CXCL10* is a key determinant of severe COVID-19[18]. Interestingly, basal levels of *CXCL9* or *CXCL10* are elevated in elderly, hypertensive, and obese individuals, who were strongly represented in our autopsy cohort (Table 1 and Supplementary Table 1) and are predisposed to severe COVID-19[19, 20].

It has been proposed that infiltrating monocytes and macrophages have a role in lung damage[4, 21]. In support of this data, we found CD68^+ macrophage infiltrates to be positively associated with DAD (Fig. 5j). In addition, DAD was associated with the activated cytotoxic T cell signature ($p$ = 0.0022) (Fig. 5k), yet not with the overall numbers of pulmonary CD8^+ T-cells (Fig. 5l). This raises the possibility that activated CD8^+ T cells contribute to DAD as they eliminate the virus from the infected lungs. None of the above pulmonary cytokine sub-signatures, however, was positively associated with DAD (Fig. 5m–p), suggesting that none of these cytokines drives lung pathology directly.

In summary, we did not find distinct features of lung damage in ISG^high patients, suggesting that extra-pulmonary factors may contribute to mortality in these patients. However, ISG^low patients show prominent DAD, associated with peri-alveolar foci of CD68^+ macrophages and an activated T cell signature. Local expression of most cytokines did not correlate to lung damage, except for *CXCL9/10/11*, which correlated to IAH ($p$ = 0.018) (Fig. 5i). Based on studies that associated serum *CXCL10* levels with general disease severity[18], it will be interesting to investigate

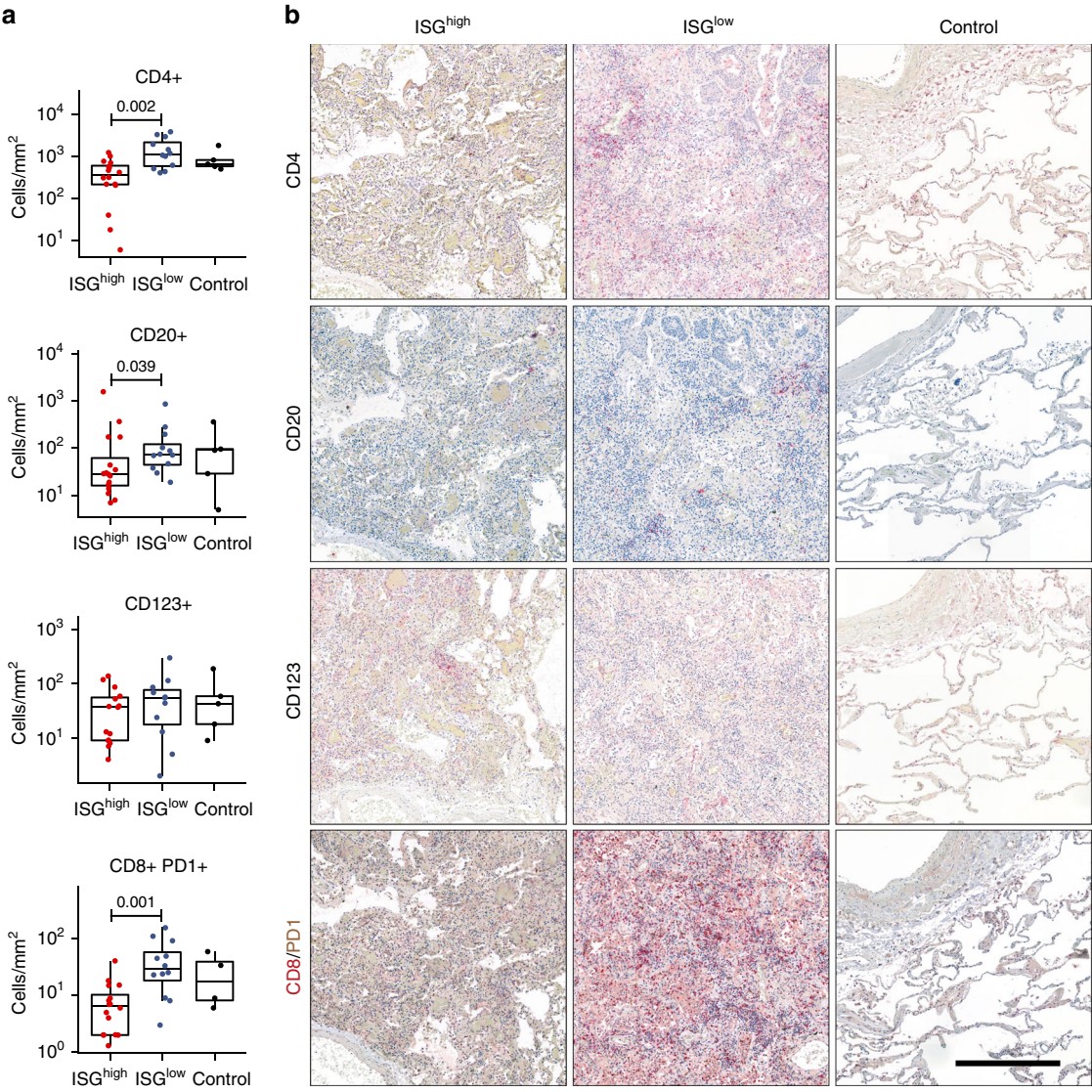

**Fig. 4 Immune cell infiltrates on COVID-19 lung sections. a** Frequencies of immune cells on ISG$^{high}$ and ISG$^{low}$ COVID-19 lung sections and controls ($n$ = 33 for CD4 and CD20, $n$ = 31 for CD123, $n$ = 32 for CD8/PD1). Box-plots elements indicate the median (center line), upper and lower quartiles (box limits). Whiskers extend to the most extreme value included in 1.5× interquartile range. Groups were compared using a two-sided Wilcoxon rank-sum test. **b** Representative immunohistochemistry (CD4, CD20, CD123, CD8/PD1) of ISG$^{high}$ and ISG$^{low}$ COVID-19 lungs and controls, size bar 500 μm. At least two different tissue blocks from different areas of the lungs were evaluated for each case. ISG$^{high}$ samples, red; ISG$^{low}$ samples, blue.

whether serum levels of these cytokines predict a specific pattern of lung damage.

**The ISG$^{low}$ lung profile shows signs of tissue regeneration and T cell exhaustion**. Since ISG$^{low}$ lung samples were derived from patients with a longer disease course, we investigated specific pathways of local immune regulation and tissue regeneration. ISG$^{low}$ lung samples expressed elevated p53 and Ki67 (Figs. 1a and 6a), i.e., reactive markers indicating lung remodeling in DAD[22].

Since we found local upregulation of *C1QA* ($p$ = 0.017) and *C1QB* ($p$ = 0.0012) specifically in ISG$^{low}$ lungs (Figs. 1a and 6b), we hypothesized that complement activation may further contribute to lung damage in these patients. Consistently, we found strong staining for C3d and C5b-9 complex deposition in lung tissue indicating complement activation in the lungs of ISG$^{low}$ patients (Fig. 6c). Since *C1Q* also restrains antiviral CD8$^{+}$ effector T cell responses[23], it may contribute to the local

regulation of effector T cells. In line with previous observations[24], we found a higher frequency of CD8$^{+}$PD1$^{+}$ T-cells in the ISG$^{low}$ subgroup ($p$ = 0.001), potentially indicative of T cell exhaustion (Fig. 4a, b).

Overall our results identify two patterns of pulmonary COVID-19 disease that lead to death from respiratory failure. Patients of the ISG$^{high}$ subgroup die early with high viral loads and high cytokine and ISG expression levels in the lungs. Their lungs are morphologically relatively intact, and our data do not identify a uniform pathomechanism underlying lethal outcome, although some show *CXCL9/10/11*-associated IAH (Fig. 5i). It is possible that the direct cytopathic effect of SARS-CoV-2 on alveolar epithelial cells may have contributed to the lethal outcome. The distinct ISG$^{low}$ group of patients dies later, with low viral loads in the lungs, low local expression of cytokines and ISGs, yet strong infiltration of pulmonary tissue by CD8$^{+}$ T cells and macrophages, which both correlate to the severity of DAD and local complement activation. Some of these patients show IAH in addition to DAD, and many of them suffer from

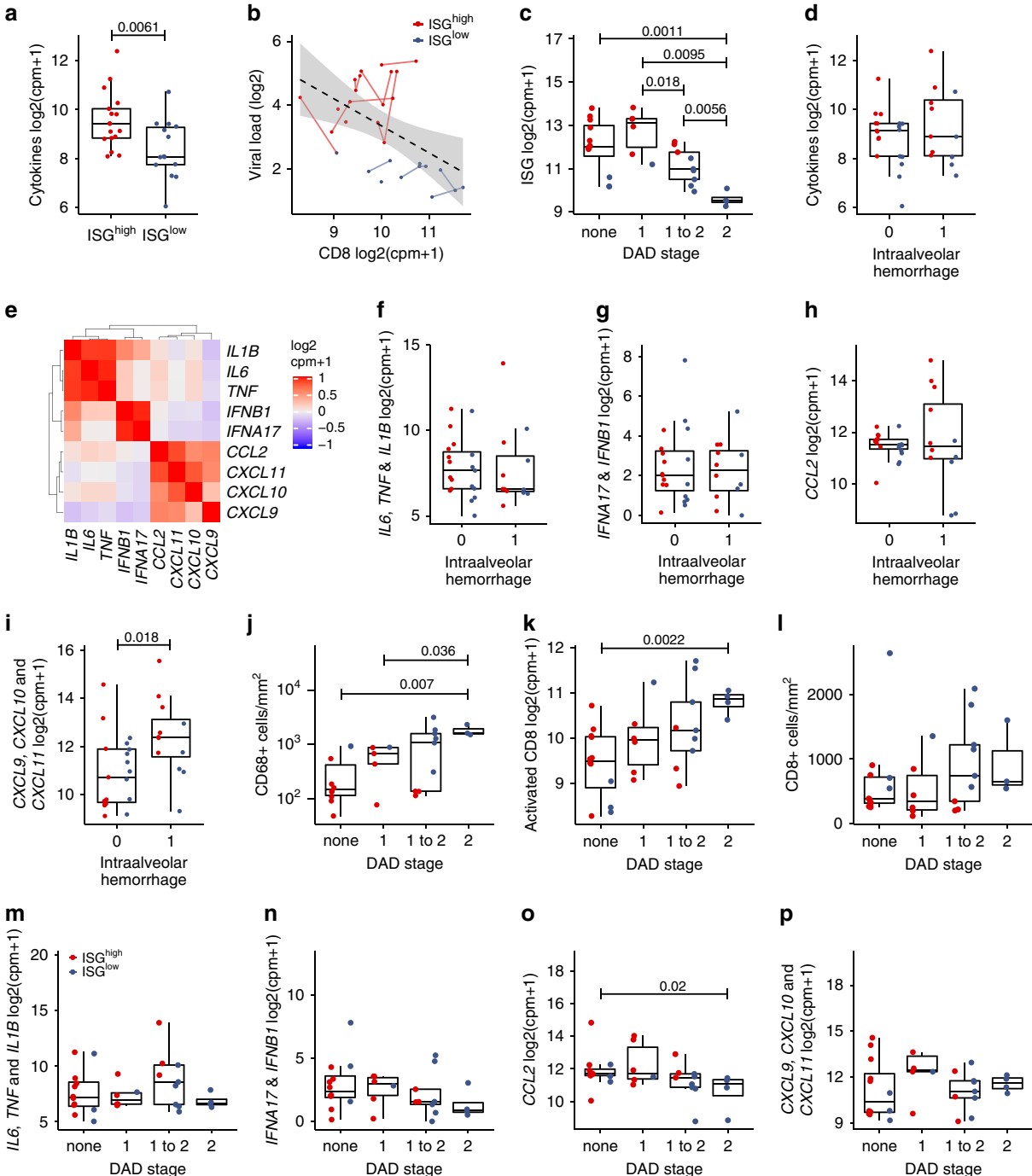

**Fig. 5 Correlation of ISG^high and ISG^low lung immunoprofiles with morphological changes. a** Expression of a cytokine signature (*TNF, IL-1B, IL6, IFNA17, IFNB1, CCL2, CXCL9, CXCL10, CXCL11*) in ISG^high and ISG^low COVID-19 lung samples. This pro-inflammatory cytokine signature was significantly enriched in the ISG^high subset (*n* = 31 independent samples). **b** Inverse correlation of viral load and activated CD8^+ T cell signature (*CD38, GZMA, GZMB, CCR5*). Solid lines connect sample points from the same patient. The dotted line shows a regression for all the samples, and the gray area delimits the 95% Confidence Intervals around it (Pearson's correlation = −0.5, adjusted *R*-squared = 0.22, *p*-value = 0.005). **c** Association of DAD stage with ISG expression (*n* = 31 independent samples). **d** Association of the pro-inflammatory cytokine signature with intra-alveolar hemorrhage (IAH) (*n* = 31 independent samples). **e** Pearson's correlation of pro-inflammatory cytokines in the cytokine signature indicates the presence of co-regulated cytokines. **f–i** Association of cytokine signatures in ISG^high and ISG^low COVID-19 lung samples with IAH. Association of: **f** Median *IL6, TNF, IL1B* expression. **g** Median *IFNA17, IFNB1* expression. **h** Median *CCL2* expression. **i** Median *CXCL9/10/11* expression in ISG^high and ISG^low COVID-19 lung samples versus IAH. Only the *CXCL9/10/11* sub-signature was positively associated with IAH (*n* = 31 independent samples). **j** Association of CD68^+ macrophage infiltrates with DAD (*n* = 27 independent samples). **k** Association of DAD stage with activated CD8^+ T cell signature (*n* = 31 independent samples), **l** with CD8^+ T cell counts (*n* = 29). **m–p** Association of cytokine signatures in ISG^high and ISG^low COVID-19 lung samples with DAD stage. Association of: **m** Median *IL6, TNF, IL1B* expression. **n** Median *IFNA17, IFNB1* expression. **o** Median *CCL2* expression. **p** Median *CXCL9/10/11* expression in ISG^high and ISG^low COVID-19 lung tissue with DAD stage (*n* = 31 independent samples). ISG^high samples, red; ISG^low samples, blue. All box-plots elements indicate the median (center line), upper and lower quartiles (box limits). Whiskers extend to the most extreme value included in 1.5× interquartile range. Groups were compared using a two-sided Wilcoxon rank-sum test.

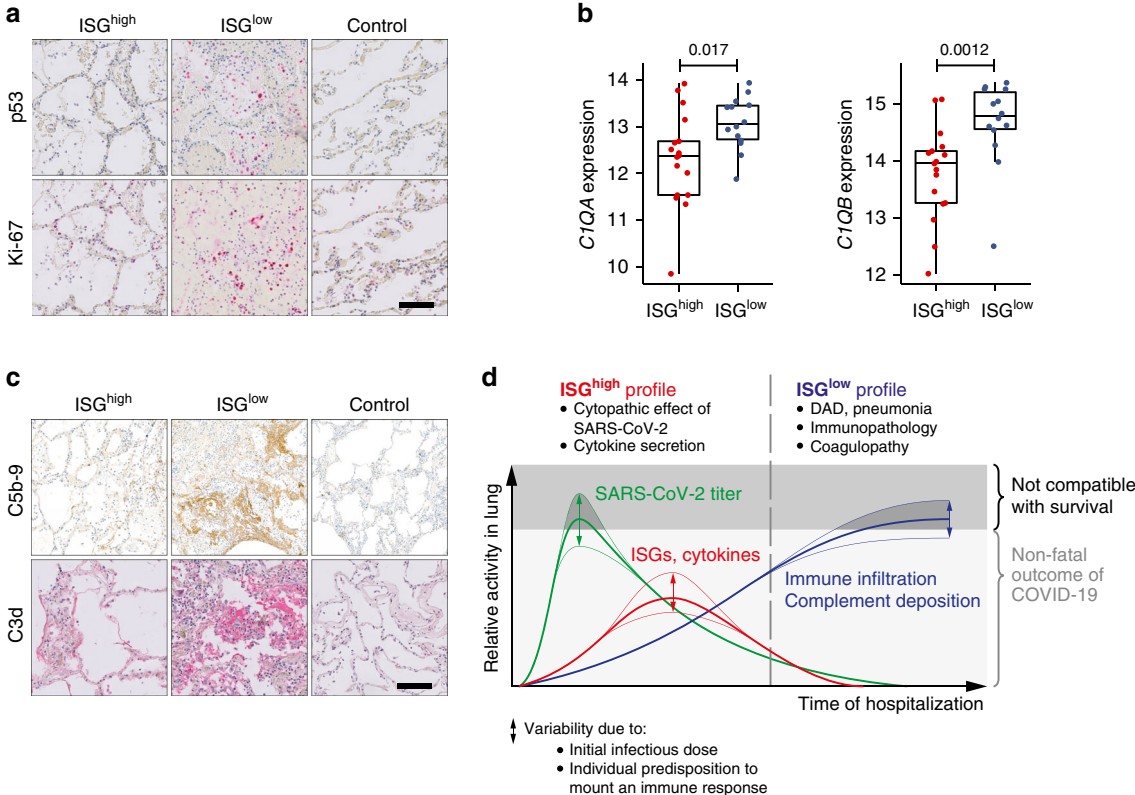

**Fig. 6 Molecular characteristics of the ISG^high and ISG^low COVID-19 lung profiles. a** Representative immunohistochemistry for p53 and Ki67. Size bar 100 μm. At least two different tissue blocks from different areas of the lungs were evaluated for each case. **b** Expression of *C1QA* and *C1QB* in ISG^high and ISG^low lung samples (*n* = 31 independent samples). Box-plots elements indicate the median (center line), upper, and lower quartiles (box limits). Whiskers extend to the most extreme value included in 1.5× interquartile range. Groups were compared using a two-sided Wilcoxon rank-sum test. **c** Representative IHC stainings for complement activation products C5b-9 and C3d in ISG^high, ISG^low COVID-19, and normal control lungs. Size bar 100 μm. At least two different tissue blocks from different areas of the lungs were evaluated for each case. ISG^high samples, red; ISG^low samples, blue. **d** Schematic time course of COVID-19 lung disease based on lung autopsy findings. Early in the disease, an ISG^high lung profile is observed, with high viral load, high expression of cytokines and ISGs, and sparse immune infiltrates. Late in the disease, an ISG^low lung profile is observed, with low viral load, low local expression of cytokines and ISGs, and strong infiltration of macrophages and lymphocytes. Patients who die early are not able to adequately control SARS-CoV-2, while patients who die late suffer from DAD and immunopathology. Infectious dose and individual predisposition to mount immune responses likely define whether or not a patient survives COVID-19. Green line: Relative viral loads, red line: Relative expression of lung ISGs and cytokines, blue line: pulmonary immune infiltrates and complement deposition. Dark gray area: lethal outcomes, arrows: individual variability.

coagulopathies. Altogether this patient group appears to suffer from severe pulmonary immunopathology. Based on the time after hospitalization, and based on knowledge about the general course of coronavirus infections[14], our data suggest that patients expressing an ISG^low profile in their lungs might have undergone a previous ISG^high phase, and it is possible that the initial infectious dose and individual predisposition defines when and whether a patient dies of COVID-19 lung disease (Fig. 6d).

## Discussion

Here we describe two immunopathological patterns in the lungs of fatal COVID-19 patients based on ISG expression. The ISG^high pattern is observed in patients, who die early after hospitalization and is characterized by high viral load and high levels of proinflammatory cytokines, yet relatively intact lung morphology, while the ISG^low pattern is characterized by low viral load, massive lung damage, marked lung immune cell infiltrates, and late death. Our findings are consistent with epidemiological data showing two peaks of mortality[25], and another study of four COVID-19 autopsies, where one patient died early after hospital admission, with striking upregulation of pulmonary IL-1b/IL-6 in lungs and little lung damage, while three patients expressed low levels of pulmonary cytokines, massive DAD and delayed death[26].

Therefore, our study allows us to propose two immunopathological stages of pulmonary COVID-19.

The segregation of autopsy lung samples from COVID-19 patients in two groups based on ISG expression contributes to our understanding of the interferon response against SARS-CoV-2. Like other coronaviruses, SARS-CoV-2 is particularly sensitive to type I interferons[8, 27]. Therefore, and similar to other coronaviruses[28], it has evolved strategies to evade the interferon response, and SARS-CoV-2 leads to relatively weak IFN-I/III release in host cells at low multiplicities of infection[8]. This initial delay of IFN-I/III production may facilitate initial virus replication in the lungs, as studies with SARS-CoV in mice have shown, and enable an eventual increase of the IFN-I response and death[14]. A similar observation was made for fatal SARS-CoV infections in humans, which were accompanied by elevated expression of ISGs[29]. Since the SARS-CoV-2 receptor ACE2 is itself an ISG on lung epithelial cells[30], virus infection and the interferon response may promote each other in this phase of the infection. This may explain the observed correlation of high ISG expression and high viral load in the lungs and the widespread presence of SARS-CoV-2 in lung epithelial cells. Together this may contribute to the fatal outcome of SARS-CoV-2 infections in the ISG^high group.

The observation of the ISG^high pattern in COVID-19 autopsy lungs seems to be at odds with initial observations that critical COVID-19 patients express on average lower ISGs in blood than patients with a milder course of disease[31]. One possible explanation is that the blood ISG status is not directly reflective of the lung. In support of this idea, BALs from severe/critical patients showed high proportions of ISG^high macrophages and high expression of *CXCL9/10/11*, *IL6*, *IL-1b*, *TNF*, *CCL2*[4], which is reflective of our ISG^high phenotype in lung autopsies. An alternative explanation for the apparent disconnect of lung and blood ISG status may come from the overall frequency of the ISG^high subtype of critical/fatal COVID-19: ISGs are highly expressed in the blood of COVID-19 patients during the early, innate phase of COVID-19[9], and we show that patients with an ISG^high status in lungs die early upon hospitalization. While the early-mortality subset accounted for 44% of all deaths in our study, epidemiological data from France[25] suggest that this early critical subset is actually smaller: only 15% of patients died early after hospitalization. This percentage is consistent with the study by Hadjadj et al.[31], which detects high ISG expression in the blood of 3/17 (18%) critical COVID-19 patients, yet this signal gets diluted in the majority of ISG^low critical cases. These two alternative explanations show how critical it will be to compare gene expression in the blood and lungs of individual patients at different times of the infection and to identify peripheral biomarkers for COVID-19 lung status.

ISG^low COVID-19 patients in our study die with classical features of DAD[32], on average 9.1 days after hospitalization. Later death compared to patients with an ISG^high pattern and progressive decline of systemic ISG expression during COVID-19[9, 33] led us to infer that the ISG^low pattern in lungs reflects a later phase of pulmonary COVID-19. ISG^low lungs show higher frequencies of T and B lymphocytes, compared to ISG^high lungs. None of our fatal cases showed lung lymphocyte counts below control levels. Therefore COVID-19-associated lymphopenia in blood[24, 34] or spleens[3] does not translate into lymphocyte depletion in infected lungs. Potential reasons are that the infected lung acts as a potent sink for circulating lymphocytes and that local proliferation limited recruitment from the blood, as was shown for CD8+ T cells in BAL of severe patients[4]. Consistent with previous observations[35], we describe an activated CD8+ T cell signature in the lungs of ISG^low patients that contain low viral counts. This suggests that CD8+ T cells are critical for antiviral protection, and may transition into a protective memory pool, as observed for SARS-CoV[36, 37]. In addition, we found elevated frequencies of CD8+PD1+ cells in ISG^low lungs compared to ISG^high lungs, but not above control levels. The observation that PD1 levels are elevated in peripheral CD8+ T cells of severe COVID-19 infection, and whether this indicates exhaustion, remains controversial[24, 33]. Overall, although we did not have paired serum antibody levels available, the infiltration pattern of ISG^low lungs suggested adaptive immune activation.

While our study sheds further light on COVID-19 lung disease, conclusions on therapy must be drawn with caution. We found that early after hospitalization, ISG^high autopsy lungs had uniformly high titers of SARS-CoV-2, and others found that viral loads in swabs and sputum are highest in early COVID-19[38]. This could indicate that treatment with compounds that directly interfere with the SARS-CoV-2 replication cycle, e.g., protease or polymerase inhibitors, should start early. However, high expression of ISGs in some lung autopsies raises caution about the use of IFN-I/III as therapeutics, at least as long as the causes and consequences of interferon signaling in COVID-19 lungs remain unclear. We found reduced viral counts in ISG^low patients but did not identify the moment at which the body is cleared of the virus. Therefore, our findings of reduced viral loads in ISG^low patients

should not be taken as justification to withhold compounds that directly interfere with the SARS-CoV-2 replication cycle from patients. Extending previous work[39], we found signs of elevated complement activation specifically in ISG^low lungs. However, it is not known whether the complement is synchronously activated in patient lungs and plasma[40]. Therefore, our results do not provide a further step toward personalized patient care but strengthen the hypothesis that complement inhibitors may show therapeutic benefit, at least in some COVID-19 patients.

Our study has several limitations. The fact that patients with the innate/ISG^high stage die early while patients with the lymphocytic/ISG^low profile die late after hospitalization, together with knowledge about the immune reaction against other coronaviruses, strongly suggests that COVID-19 lung disease progresses from an ISG^high to an ISG^low stage (Fig. 6d). However, an autopsy study is, by design, not longitudinal. Therefore, we do not have formal proof that all COVID-19 infected ISG^low lungs have undergone a previous ISG^high stage. Also, it is unknown why some patients die early and others late, and individual predisposition may be one reason. In addition, our focus on the lung only allowed us to investigate pulmonary factors of patient mortality, i.e., an overshooting innate immune activation with IAH in both ISG^high and ISG^low cases and DAD in ISG^low cases. No reported cause of death was enriched in ISG^high or ISG^low patients, and multi-organ failure was reported as a cause of death only for two of our patients. Another limitation is that we lack gene expression data from the blood of autopsy patients or serological data at the time of death. Therefore, we were not able to identify peripheral biomarkers predicting specific immunological profiles in the lung. Finally, we analyzed our lungs with a focused gene expression set since the quality and quantity of autopsy-derived RNA is often insufficient for unbiased methods. We chose to probe this gene expression dataset with pre-defined cytokine and cytotoxic T cell signatures based on published information on deregulated genes in COVID-19. This helped us put our data in perspective to the published literature, yet may have restricted our analysis. In spite of these technical limitations, we were able to uncover two novel and distinct immunopathological profiles in the lungs of fatal COVID-19.

Taken together, our autopsy study sheds light on two distinct courses of lethal COVID-19 in the lungs. It remains to be seen whether interferon signaling is only associated with or causally involved in these disease courses. However, our study strengthens the notion that interferon signaling is a central determinant of the pulmonary immune response against SARS-CoV-2.

## Methods

**Ethics statement**. This study was conducted according to the principles expressed in the Declaration of Helsinki. Ethics approval was obtained in written form from the Ethics Committee of Northwestern and Central Switzerland (Project-ID 2020-00629). For all patients, either personal and/or family consent was obtained for autopsy and sample collection, in line with Swiss law and the above Ethics approval.

**Patients and sample collection**. The study is based on the analysis of 16 out of 21 consecutive COVID-19 autopsies performed between March 9th and April 14th 2020 at the Institute of Pathology Liestal and Institute of Medical Genetics and Pathology Basel, Switzerland. Clinical features including symptoms, course of the disease, comorbidities, laboratory results, and therapy are listed in Table 1a and Supplementary Table 1. Detailed autopsy findings for each patient were recently published, and the identifiers (with the prefix "C") for each COVID-19 patient are consistent with the description of this Swiss COVID-19 autopsy cohort[6]. In this study, we analyzed formalin-fixed and paraffin-embedded (FFPE) lung tissue of distinct areas of the lungs of 16 of these COVID-19 patients. All 16 COVID-19 patients had positive nasopharyngeal swabs collected while alive. In all COVID-19 patients, the diagnosis was confirmed by detection of SARS-CoV-2 in postmortal lung tissues. 5/16 patients were additionally tested by postmortal nasopharyngeal swabs which were positive for SARS-CoV-2 in all 5 cases.

As a control cohort, we selected 6 autopsies performed between January 2019 and March 2020 at the Institute of Pathology Liestal ("normal" patients N1–N6). These control patients died of other, non-infectious causes and had similar age, gender, and cardiovascular risk profile. Patients with infections were excluded from this control cohort. Another control cohort consisted of 4 autopsies of patients suffering from various infections mainly with bacteria affecting the lung (patients with lung pathology, P1–P4). Details for both control cohorts are listed in Table 1. SARS-CoV-2 was ruled out for each control patient by PCR-examination of lung tissue samples.

**Nucleic acid extraction**. RNA was extracted from up to six sections of FFPE tissue blocks using RecoverAll Total Nucleic Acid Isolation Kit (Cat No. AM1975, ThermoFisher Scientific, Waltham, MA, USA). Extraction of DNA from up to 10 sections of FFPE tissue samples was automated by EZ1 Advanced XL (Qiagen, Hilden, Germany) using the EZ1 DNA Tissue Kit (Cat No. 953034, Qiagen, Hilden, Germany). The concentration of DNA and RNA were measured with Qubit 2.0 Fluorometer and Qubit dsDNA HS Assay or Qubit RNA HS Assay (Cat Nos. Q33230 & Q32852, ThermoFisher Scientific, Waltham, MA, USA), respectively.

**Quantification of SARS-CoV-2 in FFPE tissue samples**. Post mortem viral load was individually measured in all lung tissue blocks from all patients included in this study. SARS-CoV-2 was detected in 15 ng of human total RNA using the TaqMan 2019-nCoV Assay Kit v1 (Cat No. A47532, ThermoFisher Scientific, Waltham, MA, USA), which targets three genomic regions (ORFab1, S Protein, N Protein) specific for SARS-CoV-2 and the human RNase P gene (RPPH1). The copy numbers of the SARS-CoV-2 viral genome were determined by utilizing the TaqMan 2019-nCoV Control Kit v1 (Cat No. A47533, ThermoFisher Scientific, Waltham, MA, USA) and a comparative "$\Delta\Delta C_\tau$" method. The control kit contains a synthetic sample with a defined amount of target molecules for the human RPPH1 and the three SARS-CoV-2 assays and was re-analyzed in parallel with patient samples. For each patient sample, this method resulted in individual copy numbers of the human RPPH1 and the three SARS-CoV-2 targets. Finally, the mean copy number of the SARS-CoV-2 targets was normalized to $1 \times 10^6$ RPPH1 transcripts.

**Profiling of immune response by targeted RNAseq**. The expression levels of 398 genes, including genes relevant in innate and adaptive immune response and housekeeping genes for normalization, were analyzed with the Oncomine Immune Response Research Assay (OIRRA, Cat No. A32881, ThermoFisher Scientific, Waltham, MA, USA). The OIRRA is a targeted gene expression assay designed for the Ion™ next-generation sequencing (NGS) platform. Our study focused on the analysis of rare autopsy tissue samples from COVID-19 patients collected in clinical routine during the COVID-19 pandemic. An inherent problem for transcriptomic studies of autopsy tissues is that it is often not possible to extract high-quality RNA in sufficient amounts. To avoid sample dropout due to these reasons, we decided to use a robust and straightforward targeted gene expression assay (OIRRA) rather than whole transcriptome analysis. Since the focus of our study was to investigate the immune profile of the lungs, an immunoprofiling assay was deemed most appropriate. The OIRRA gene expression assay was originally designed to interrogate the tumor microenvironment to enable mechanistic studies and identification of predictive biomarkers for immunotherapy in cancer. The assay is optimized to measure the expression of genes involved in immune cell interactions and signaling, including genes expressed at low levels and involved in inflammatory signaling. The 398 genes covered by this assay are listed in Supplementary Table 2. The accessibility of such commercially available assays could be an encouragement to hospitals around the world to conduct similar molecular profiling studies of diagnostic tissue samples from COVID-19 patients, allowing relatively fast and easy stratification of patients into distinct biological groups as a starting point for targeted intervention strategies.

The NGS libraries were prepared as recommended by the supplier. In brief, 30 ng of total RNA were used for reverse transcription (SuperScript VILO, Cat No. 11754250, ThermoFisher Scientific, Waltham, MA, USA) and subsequent library preparation. The libraries were quantified (Ion Library TaqMan Quantitation Kit, Cat No. 4468802, ThermoFisher Scientific, Waltham, MA, USA), equimolarly pooled, and sequenced utilizing the Ion GeneStudio S5xl (ThermoFisher Scientific, Waltham, MA, USA). De-multiplexing and gene expression level quantification were performed with the standard setting of the ImmuneResponseRNA plugin (version 5.12.0.1) within the Torrent Suite (version 5.12.1), provided as part of the OIRRA by ThermoFisher Scientific, Waltham, MA, USA.

**Detection of co-infections by whole genome sequencing**. To identify potential pathogens accompanying an infection with SARS-CoV-2, we analyzed the DNA of the same tissue samples used for detection and profiling of the SARS-CoV-2-specific immune response. First, 250 ng of genomic DNA was enzymatically sheared (15 min at 37 °C) and barcoded using the Ion Xpress Plus Fragment Library Kit (Cat No. 4471269, ThermoFisher Scientific, Waltham, MA, USA). Subsequently, the libraries were quantified (Ion Library TaqMan Quantitation Kit, Cat No. 4468802, ThermoFisher Scientific, Waltham, MA, USA) and up to three libraries were pooled at equimolar levels for analysis with Ion GeneStudio S5xl

(ThermoFisher Scientific, Waltham, MA, USA). Sequencing data for each sample was analyzed using the CLC genomics workbench (version 20.0.3, Qiagen, Hilden, Germany) in combination with the microbial genomics module (version 20.0.1, Qiagen, Hilden, Germany): the raw reads were trimmed by quality (Mott algorithm with limit 0.05 and a maximum of 2 ambiguous bases per read) and mapped to the human genome (GRCh37 hg19, match score: 1, mismatch cost: 2, indel opening cost: 6, indel extension cost: 1). Unmapped reads were analyzed by taxonomic profiling to identify reads of viral or bacterial origin. The profiling utilized an index of 11,540 viral genomes with a minimum length of 1000 bp and 2715 bacterial reference genomes with a minimum length of 500,000 bp, retrieved from the NCBI Reference Sequence Database (https://www.ncbi.nlm.nih.gov/refseq/; date of download: 2020-04-02).

**Immunohistochemistry**. Immunohistochemical analyses for CD3, CD4, CD8, CD15, CD20, CD68, CD123, CD163, PD1, MPO, p53, Ki67, C3d, and C5b-9 were performed on all lung tissue blocks used in this study. Antibody stainings are part of the diagnostic routine at our institution, i.e., negative and positive controls have been established and validated on tonsil sections for all antibodies used, including isotype controls for monoclonals. All validation protocols are documented in accreditation protocol ISO 15189:2012 of our institution and are available upon request. Antibodies, staining protocols, and conditions are detailed in Supplementary Table 4.

**Qualitative and semiquantitative assessment of histopathological lung damage and neutrophilic infiltration**. Hematoxylin and eosin (H&E) and Elastica van Gieson (EvG) stained sections of all lung tissues used in this study were independently evaluated by two experienced and board-certified pathologists (VZ and KDM) (Supplementary Table 5). Both pathologists evaluated the presence of diffuse alveolar damage (DAD), and if present, its stage, intra-alveolar edema, and hemorrhage. The characteristic three phases or stages of DAD—exudative (1), proliferative/organizing (2), fibrotic (3)—were assessed as described[41]. In our cohort of COVID-19 lungs, we observed only DAD stages 1 and/or 2, and the fibrotic phase (3) was not observed. In addition, both pathologists evaluated the severity of histopathological changes in COVID-19 lungs (1 = mild/discrete alterations, 2 = moderate, 3 = severe changes) based on the resemblance between normal and pathologically altered lung tissues. Parameters that were taken into account included reduction of alveolar air-filled spaces, typical histologic features of DAD with hyaline membrane formation, infiltration of lymphocytes, monocytes, and neutrophils into interstitial and alveolar spaces, type 2 pneumocyte hyperplasia, desquamation of pneumocytes, histologic features of organizing pneumonia including intra-alveolar fibrin deposition and fibrosis (acute fibrinous and organizing pneumonia, AFOP)[42, 43]. The number of neutrophils per lung tissue section was estimated on H&E stained sections and by immunohistochemical stains for CD15 and MPO using a three-tiered system (1 = few or no neutrophils, 2 = moderate number of neutrophils, 3 = high number of neutrophils). Assessment of the two pathologists was concordant in the vast majority of cases. Discrepant cases were reviewed by a third pathologist (NW) to reach consent.

**Digital image analysis**. Slides were digitalized on a 3DHistech™ P1000 slide scanner at ×400 magnification (3DHISTECH Ltd. Budapest, Hungary). Digital slide review and quality control were performed by a board-certified pathologist (VHK). Tissue regions with staining artefacts, folds, or other technical artefacts were excluded from the analysis. A deep neural network (DNN) algorithm (Simoyan and Zisserman VGG, HALO AI™ on HALO™ 3.0.311.167, Indica Labs, Corrales, NM) was trained using pathologist annotations to automatically localize and measure the area of each lung tissue sample on the digital slides. Background regions and glass were excluded from the analysis. Mark-up images for tissue classification were generated and classification accuracy was confirmed through pathology review. For cell-level analysis, color deconvolution for DAB, AP, and hematoxylin channels was performed and nuclear segmentation was optimized using cell-morphometric parameters. Marker-positive cells in stromal and epithelial regions were quantified. For CD3, CD4, CD8, CD20, CD68, CD123, CD163, and PD1, staining detection was optimized for the cytoplasmic/membranous compartment, and marker expression was measured on a continuous scale at single-cell resolution. For assessment of CD8/PD1 double stains, color deconvolution was optimized for the separation of DAB (PD1) and AP (CD8) staining products. Internal controls (non-immune cells) and external controls (tonsil) were used to calibrate the detection limits and cross-validated by visual review. For each tissue sample, the total area of lung tissue in mm², the absolute number of marker-positive cells, cell-morphometric parameters, and staining intensity were recorded.

**Identification of SARS-CoV-2 immune response pattern**
*Gene expression analysis*. Samples were included in the study based on the quality of libraries and alignment performance. Applied inclusion criteria are >1 million mapped reads, good concentration of libraries, average read length >100 bp, >300 target genes with more than 10 reads. One sample with >1 Mio reads was excluded from the study because of shorter read length and a low library concentration. Notably, this sample had the longest time between death and autopsy (72 h) before analysis. Differential expression analysis was performed using the edgeR package comparing normal lung samples, COVID-19 samples, and samples from patients

with other infections. Genes were selected for downstream analyses by fdr <0.05 and |logFC| > 1 for clustering analysis. Clustering analysis was performed using $k$-means algorithm and complete linkage. The ideal number of clusters ($n = 3$) was chosen based on 30 different algorithms[44] and the final clustering derives from the consensus of 2000 iterations. The expression of gene signatures was calculated as the median of log2(cpm + 1) of selected genes.

*Functional enrichment analysis.* Biological processes enrichment was performed using the enrichGO function of the package clusterProfiler[45] setting all the genes included in the assay as universe.

*Statistical analysis and reproducibility.* All the analyses and graphical representations were performed using the R statistical environment software[46] and the following packages: ggplot2[47], circlize[48], ComplexHeatmap[49], ggfortify[50], reshape2[51], and factoextra[52]. Correlation between transcripts and viral counts was performed using Pearson's correlation. Association between continuous and categorical data were tested using the Wilcoxon rank-sum test.

Based on the often non-uniform histopathological appearance of lung samples from the same patient, transcriptomic, morphologic, or histopathological analyses were performed at the tissue sample level. Analyses involving patients' clinical or demographical data were performed at the patient level and patients in which all analyzed lung samples expressed an ISG^high or an ISG^low profile were called ISG^high or ISG^low patients.

Box-plots elements indicate the median (center line), upper and lower quartiles (box limits), and show all the data points. Whiskers extend to the most extreme value included in 1.5× interquartile range.

The microscopic images are representative of at least two different tissue blocks from different areas of the lungs that were evaluated for each COVID-19 case. All data are representative of 34 post mortem lung samples from 16 deceased COVID-19 patients. At least two different tissue blocks from different areas of the lungs were evaluated for each case. Results were compared to nine post mortem lung samples from six patients who died from non-infectious causes.

**Reporting summary**. Further information on research design is available in the Nature Research Reporting Summary linked to this article.

## Data availability

The data sets generated and analyzed within this study can be accessed in GEO (GSE151764). Patients' clinical features are listed in Table 1 and Supplementary Table 1. Source data are provided with this paper.

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

## Acknowledgements
V.H.K. gratefully acknowledges funding by the Promedica Foundation (F-87701-41-01). A. T., J.D.H., T.M. and K.D.M. are supported by and gratefully acknowledge the Botnar Research Centre for Child Health. We would like to thank Christian Tosch, Beat Béni, Daniel Turek, Melanie Sachs, Anne Graber, Christina Herz, Arbeneshe Berisha, Norbert Wey and the USZ pathology IT team, André Fitsche, Marcel Glönkler, Christiane Mittman and the USZ pathology laboratory team for expert technical support and scanning of slides.

## Author contributions
R.N., Y.C., V.H.K., F.D., T.J., and K.D.M. jointly conceived the study, performed data interpretation, and prepared the manuscript. A.T., M.B., H.M., M.T., and C.A. provided intellectual input, provided critical resources, and critically reviewed the manuscript. R.N., Y.C., V.H.K., M.H., T.H., and T.J. performed the bioinformatic and statistical analysis. A.T., J.D.H., T.M., N.S., A.F. collected autopsy specimens, patient data, and performed experiments. V.Z., N.W., W.K., and C.A. performed the histomorphological evaluation. All authors approved the final manuscript.

## Competing interests
V.H.K. has served as an invited speaker on behalf of Indica Labs. T.H. and T.J. are employees of Novartis. The remaining authors declare no competing interests.
