## [Peer Review File · Nature Communications]

REVIEWERS' COMMENTS

Reviewer #2 (Remarks to the Author):

The authors have address the major comments with additional clarification and discussion. The discussion, in particular, is an important addition to this study, as it brings up important comparisons and concepts that the data bring up with regard to the role of Type I IFNs in anti-viral immunity and immune pathology.

Reviewer #4 (Remarks to the Author):

I have no further comments/objections

Reviewer #5 (Remarks to the Author):

1. The authors seem to leave open two interpretations.

Lines 126-133 summarise the transcriptomic and histopathological data as identifying 'two distinct clinical courses of fatal COVID-19', namely ISGhi and ISGlow. However, the final sentence of this paragraph states 'the ISGhi profile precedes the ISGlow'. Given the differences in disease duration between these groups, the latter interpretation seems most likely. If so, this does not support the conclusion that there are 'distinct clinical courses.' It seems simply that ISGhi cases die earlier in disease but might otherwise progress to ISGlow. Do the authors agree? If they do, this emphasis throughout the manuscript would make the conclusion clearer.

However, I do accept the point they make that death provides a once-only snapshot of events as they are at a particular time and that sequential evolution of different stages of disease are hard to infer from post-mortem studies.

2. In the paragraph on 'cytokine storm' (lines 181-190), the authors present only 4 patients with non-COVID infections (of mixed viral and bacterial aetiology). It seems that nothing specific to COVID-19 can be concluded given this very small number of controls. Reference 4 in this section also does not include data from non-COVID controls. This is essential to defining a 'cytokine storm' versus simply a proportionate inflammatory immune response. The evolving view is that COVID is not really a 'storm'.

3. The manuscript relies on several instances of co-expressed transcript sets (e.g. figure 5) or signatures based on a selection of genes (e.g. the cytotoxic T cell signature). The way these were selected is not apparent. Additionally, the selection of some gene sets (e.g. the inflammatory or CXCL modules in figure 5 and combination of data seems to artificially increase difference between groups. I suggest that if the authors wish to take a modular approach to the analysis, then a formal analysis using e.g. GO or KEGG terms would be more valid.

4. I would like to see a table demonstrating the apparent evolution of disease, or a visual summary in the form of a diagram showing how the findings fit together

Re: NCOMMS-20-25384-A

Two distinct immunopathological profiles in autopsy lungs of COVID-19

Point-by-point response to Reviewers' comments

Reviewer #2

(Remarks to the Author):

The authors have addressed the major comments with additional clarification and discussion. The discussion, in particular, is an important addition to this study, as it brings up important comparisons and concepts that the data bring up with regard to the role of Type I IFNs in anti-viral immunity and immune pathology.

We thank the reviewer for her / his interest in our work and for her / his input and comments.

Reviewer #4

(Remarks to the Author):

I have no further comments/objections.

We thank the reviewer for her / his interest in our work and for her / his input and comments.

Reviewer #5

(Remarks to the Author):

1. The authors seem to leave open two interpretations.

Lines 126-133 summarise the transcriptomic and histopathological data as identifying 'two distinct clinical courses of fatal COVID-19', namely ISGhi and ISGlow. However, the final sentence of this paragraph states 'the ISGhi profile precedes the ISGlow'. Given the differences in disease duration between these groups, the latter interpretation seems most likely. If so, this does not support the conclusion that there are 'distinct clinical courses.' It seems simply that ISGhi cases die earlier in disease but might otherwise progress to ISGlow. Do the authors agree? If they do, this emphasis throughout the manuscript would make the conclusion clearer.

However, I do accept the point they make that death provides a once-only snapshot of events as they are at a particular time and that sequential evolution of different stages of disease are hard to infer from post-mortem studies.

We thank the reviewer for her / his interest in our work and for her / his input and comments.

We were very cautious with the interpretation of our *post-mortem* data, and we appreciate that this may have led to ambiguities. And we do agree with the reviewer that our data on hospitalization times suggest that ISGhigh and ISGlow are actually different stages of COVID-19 lung disease. Moreover, there is evidence from SARS-CoV-1 showing that IFN signaling precedes lethal pneumonia (Channapanavar, Cell Host Microbe 2016). Based on this evidence, and based on the above comment by the reviewer, we now provided this likely interpretation of our data throughout the manuscript – while still mentioning the obvious limitation of *post-mortem* studies in the Discussion. In addition we further illustrated this interpretation by a visual summary (**Figure 6d**) as suggested in Point 4 of this Reviewer.

2. In the paragraph on ‘cytokine storm’ (lines 181-190), the authors present only 4 patients with non-COVID infections (of mixed viral and bacterial aetiology). It seems that nothing specific to COVID-19 can be concluded given this very small number of controls. Reference 4 in this section also does not include data from non-COVID controls. This is essential to defining a ‘cytokine storm’ versus simply a proportionate inflammatory immune response. The evolving view is that COVID is not really a ‘storm’.

Thanks for pointing this out. The initial version of the manuscript was written at a time where a “cytokine storm” was seen as the most likely root cause of COVID-19 mortality, and this view is gradually changing. Therefore we have now replaced the increasingly contentious term of “cytokine storm” with a more factual terminology.

3. The manuscript relies on several instances of co-expressed transcript sets (e.g. figure 5) or signatures based on a selection of genes (e.g. the cytotoxic T cell signature). The way these were selected is not apparent. Additionally, the selection of some gene sets (e.g. the inflammatory or CXCL modules in figure 5 and combination of data seems to artificially increase difference between groups. I suggest that if the authors wish to take a modular approach to the analysis, then a formal analysis using e.g. GO or KEGG terms would be more valid.

Thanks for this question. Our CTL signature was assembled based on genes, which are known to be associated with severe COVID-19. For example, CD38+CD8+ cells are specifically enriched in severe patients (e.g. Song, Nat Comms 2020) and GZMA/B and CCR5 are strongly elevated in CTL from critical patients (e.g. Chua, Nat Biotechnology 2020). The co-expression analysis of cytokines, i.e. the generation of cytokine modules in Figure 5e was done in an unbiased fashion, yet the choice of cytokines to include in this analysis was again based on prior evidence. For example, CXCL9/10/11 are consistently upregulated in BAL of COVID-19 patients (Liao, Nat Med 2020); IL-1b, IL-6 and TNFa are upregulated in plasma of COVID-19 patients (Huang, Lancet 2020), and IFNa/b are dysregulated following SARS-CoV-2 infection (Blanco-Melo, Cell 2020). We recognize that selection of a gene signature based on prior evidence and the limited number of genes in our analysis may potentially lead to a bias in our analysis, and we now point this out specifically in our Discussion.

4. I would like to see a table demonstrating the apparent evolution of disease, or a visual summary in the form of a diagram showing how the findings fit together.

As stated in reply to point 1 by the reviewer, we now provide a graphical summary to the article (**Figure 6d**) to help illustrate our data and their interpretation.